# Federated Link Prediction on Dynamic Graphs

## Abstract

Link prediction on dynamic, large-scale graphs has been widely used in real-world applications, such as forecasting customer visits to restaurants or predicting product purchases. However, graph data is often localized due to privacy and efficiency concerns. Training separate local models based on data in each region preserves privacy but often leads to less accurate models, especially in smaller regions with fewer users and products. Federated learning then collaboratively trains models on localized data to maintain model accuracy and data privacy. However, the vanilla FL approach requires training the entire historical graph of user interactions, introducing high computational costs during training. While training on the most recent data may help reduce overhead, it decreases the model accuracy and incurs data imbalance across clients. For instance, regions with more users will contribute more training data, potentially biasing the model toward those users. We introduce FedLink, a federated graph training framework for solving link prediction tasks on dynamic graphs. By continuously training on fixed-size buffers of client data, we can significantly reduce the computation overhead compared to training on the entire historical graph, while still training a global model across regions. Experiments demonstrate that FedLink matches the accuracy of training a centralized model while requiring $3.41\times$ less memory and running $28.9\%$ faster compared with full-batch federated graph training.

## 1 Introduction

Dynamic graphs have been widely applied in recent years, e.g., recommendations and advertisements by predicting customer visits to restaurants and user purchases of products (Kazemi et al., 2020). In such applications, we represent users and items (i.e., restaurants or products) as nodes in a graph, with links between them representing purchases and node features representing characteristics of the users and items that are relevant to their consumption behavior (Figure 1 left). This graph-based formulation allows us to exploit recent advances in Graph Neural Networks (GNNs) for predicting the links between users and items to make and update accurate predictions of user behavior (Zhang & Chen, 2018; Chen et al., 2022; Guo et al., 2023; Wang et al., 2023; Huang et al., 2024). Moreover, by representing user data as a dynamic graph that evolves as new user data is collected, e.g., as users visit new restaurants and new users engage with a recommendation platform, we can capture temporal information that static graphs cannot collect and make more up-to-date predictions (Pareja et al., 2020; Yu et al., 2023; Huang et al., 2023; You et al., 2022; Cong et al., 2023). However, recent trends toward data localization introduce new challenges in applying these dynamic GNN models across different regions.

In practice, applying GNNs to dynamic graph recommendation data faces **practical challenges**. Instead of training region-specific models only based on the local data, we would like to develop a *unified GNN model that trains on all regions' data and applies to customers in all business regions*. Such a centralized model is often especially helpful for small regions with limited local data, since users in different regions may exhibit similar patterns and this model can thus benefit from data collected in more populated regions. Users may also move from one region to another, e.g., Figure 1 (upper right) illustrates a user moving from Region A to Region B. Thus, region-specific models likely will not generalize well to such users: link recommendation models generally rely on generating user and link embeddings, and users' embeddings from their old regions may no longer generate accurate predictions in their new regions if the trained models are region-specific.

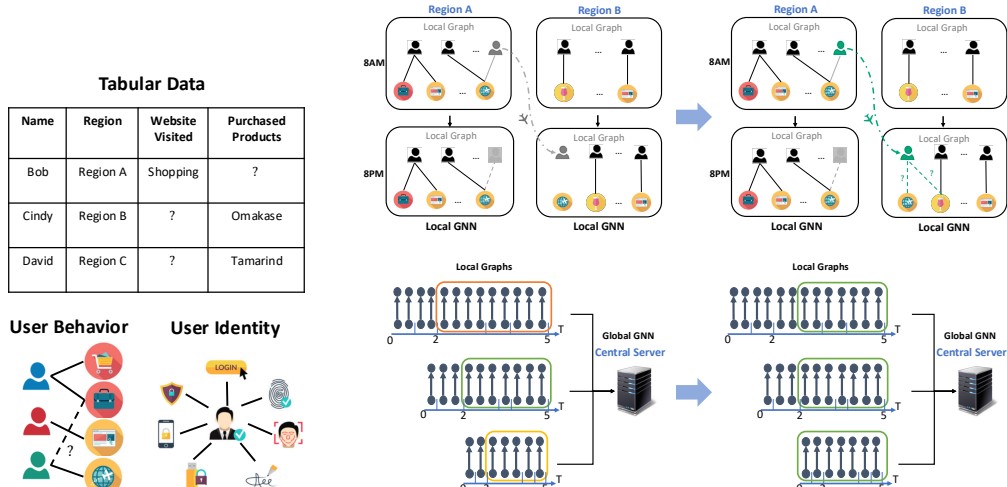

Figure 1: (Left) Three examples use cases of federated learning on dynamic graphs; (Upper Right) Sample architecture for federated training on local dynamic graphs in different regions where: 1. Local graphs are heterogeneous and differ in their size comparing different regions; 2. Clients can have temporal edges in different timestamps; 3. Originally, user (gray) who travel from one region to another will not carry their historical link information to prevent privacy leakage, thus requiring users (green) to predict their behavior in the new region; (Lower Right) Illustration of temporal imbalance within and across clients, from uniform time-interval sampling showing varying node counts per client on the left transitioning to a constant edge buffer approach on the right.

.

Training a centralized GNN model, however, is more and more constrained by *strict data protection regulations* such as Payment Aggregators and Payment Gateways (PAPG) in India and General Data Protection Regulation (GDPR) in Europe (EU), which generally forbid data from being sent out of a region. Even without these regulations, centralized training with billions of graph nodes at clients across the world incurs *high computation costs*, (e.g. billions of users and places in Facebook) (Ching et al., 2015). Federated learning (Kairouz et al., 2021; Tan et al., 2022; Ghosh et al., 2020; Deng et al., 2020; Zhou et al., 2021) helps meet these challenges by allowing servers in different geographic regions to maintain local prediction models that are periodically synchronized with a global model at a central server, thus keeping data at the locations where it is generated. However, *federated learning itself does not alleviate the computational overhead at each client (region)*, which may need to train over millions of users and items in each training round.

Vanilla implementations of federated learning (e.g., STFL (Lou et al., 2021) and FedGraphnn (He et al., 2021)) in our dynamic graph setting would train on the entire historical graph in each federated learning iteration. However, doing so incurs *high training overhead*, as the historical graph may consist of billions of links (e.g., for regions with millions of users who have visited millions of restaurants over the past few months), which only grows over time. Training on the entire graph then requires enormous amounts of both memory and training time. Reducing the memory and training time by training on only the recent graph snapshot (e.g. a few days of data by 4D-FED-GNN+ (Gürler & Rekik, 2022)), however, limits model accuracy as it only incorporates recent information on user preferences.

Other past works on federated graph learning (Wang et al., 2020; Yuan et al., 2022) iteratively train on a time series of graphs for each region, e.g., based on snapshots taken every hour, day, etc. However, in practice, many regions' graphs may have only a few new links in each time slot (Jin et al., 2022), while some regions, particularly those with larger populations, may have a vast amount of link interactions, as shown in Figure 1 (lower right). Thus, restricting each client to only train on edges arriving in the last timeslot will introduce a new challenge of *spatial and temporal data heterogeneity* across clients. Such client heterogeneity

is known to hinder the convergence of federated learning algorithms (Ye et al., 2023), as it can skew the trained model towards clients with more training data in any given round, at the expense of other clients.

To overcome the challenges of data heterogeneity and large training overhead, we realize that ***maintaining buffers to store the same number of previously arrived edges at each client*** will dramatically reduce the computation and memory cost for the training, solving our computational overhead challenge, while also naturally including the same number of new links at each client in the training, solving our heterogeneity challenge. Figure 1 (lower right) illustrates this idea. By enabling efficient federated training, this idea also ensures that our learned models will be more accurate for users moving between regions: users can simply transfer their learned embeddings between regions, with minimal re-training needed when they move from one region to another. We use this insight to make the following **contributions:**

- We introduce **FedLink**, a federated GNN training framework for solving link prediction tasks on dynamic graphs. FedLink significantly reduces training overhead by continuously training on historical data buffers of the same size at each client, balancing data heterogeneity across different clients (regions).

- We **theoretically analyze** and **empirically validate** the effect of buffer size on model performance and indicate the tradeoffs between staleness, if a large buffer with possibly outdated data is used, and dataset size, as well as the increased memory and training time with larger buffers.

- **Experiments** on link prediction across multiple regions demonstrate that FedLink significantly reduces GPU memory usage (by up to $3.41\times$) and training time (by up to $28.9\%$) compared to full-batch federated graph training, with equivalent prediction accuracy.

We give an overview of related work in Section 2 before introducing our FedLink algorithm, along with its theoretical foundation and empirical validation in Section 3. We then present extensive distributed experiments to demonstrate FedLink's performance on distinct subsets of the FourSquare dataset, which contains 22,809,624 check-ins in 77 countries, in Section 4 and conclude in Section 5.

## 2 Related Work

**Graph neural networks** aim to learn representations of graph-structured data (Bronstein et al., 2017). GCNs (graph convolutional networks) (Kipf & Welling, 2016), GraphSage (Hamilton et al., 2017), and GAT (graph attention network) (Veličković et al., 2017), for example, have shown excellent performance on various graph learning tasks like classifying nodes based on their and their neighbors' features. Dynamic graphs provide temporal information on the links between nodes to further improve the task performance. Many methods are proposed for training GNNs on dynamic graphs with recurrent structure (Chen et al., 2022; Pareja et al., 2020; Yu et al., 2023) or with specific training methods (Huang et al., 2023; You et al., 2022; Cong et al., 2023). Given such dynamic graphs, a popular task is that of link prediction (Zhang & Chen, 2018), i.e., predicting whether a link will be formed between two nodes in the graph, given information on the presence or absence of links between other pairs of nodes.

Recently, federated learning (Kairouz et al., 2021; Tan et al., 2022; Ghosh et al., 2020; Deng et al., 2020; Zhou et al., 2021) has become popular for communication efficient multi-device training with privacy preservation. Many works have proposed methods to train GNN models on static graphs in a federated learning setting (Liu et al., 2024; Wang et al., 2022a), e.g. GCFL (Xie et al., 2021), FedSage+ (Zhang et al., 2021b), FedGCN (Yao et al., 2024a), and FedPub (Baek et al., 2023). The development of federated graph learning libraries has also accelerated research progress in this area (Yao et al., 2024b). Moreover, several methods are proposed for training GNNs on **dynamic graphs in the federated learning** setting, driven by the plethora of applications that can be modeled with federated dynamic graphs. However, such works are mainly application-specific, in particular considering traffic flow forecasting and trajectory prediction. For example, CNFGNN (Meng et al., 2021) provides a spatio-temporal model for a cross-node federated GNN, where each client is a node in the graph, validating their algorithm in a traffic flow forecasting application. ATPFL (Wang et al., 2022b) combines Automated Machine Learning (AutoML) with federated learning for federated multi-source trajectory prediction. Federated Community GCN (Xia et al., 2022), Spatial-Temporal Long and

Short-Term Networks (FedSTN) (Yuan et al., 2022), and Attention-based Spatial-Temporal Graph Neural Networks (ASTGNN) (Zhang et al., 2021a) have all proposed for traffic speed forecasting with FL. However, these works are specific to traffic flow forecasting and trajectory prediction applications and thus cannot be directly applied to more general problems or recommendation scenarios.

More generally, STFL (Lou et al., 2021) provides a spatio-temporal model for federated graph classification, where the task is to classify the label of each client's graph. Feddy (Jiang et al., 2022) considers temporal information in the graph embedding by applying dynamic GNNs, while 4D-FED-GNN+ (Gürler & Rekik, 2022) focuses on the evolution graph learning task with missing time points. Each client trains a GNN model on the graph constructed at the current time step. None of these works, however, explicitly consider imbalances in data across clients or time, as we do in this work. Moreover, training on the entire historical graph, as in STFL, can introduce significant computation overhead with potentially little gain: links formed a long time ago may no longer be useful in predicting new links, e.g., if user tastes change over time. To the best of our knowledge, *FedLink is the first framework and algorithm for federated link prediction on dynamic graphs that tackles these challenges*, and we compare its performance to these baselines in Section 4.

## 3 FedLink

In this section, we first formalize our federated link prediction problem and introduce our FedLink training algorithm. We then present a theoretical rationale and empirical validation for FedLink's buffer-based design.

### 3.1 Federated Link Prediction

**Federated graph setup.** Suppose we have $N$ users and $M$ items. We consider $K$ clients each located in a different geographic region, e.g., different countries, and one central server to coordinate among the clients. We use $i = 1, 2, \ldots, I_k$ and $j = 1, 2, \ldots, J_k$ to respectively index users and items within each region $k$; while users may move from one region to another, items do not (e.g., if the items are restaurants physically located in specific regions). These users and items comprise the nodes of client $k$'s local graph. Each client $k$ receives a stream of link interactions (user-item interactions) occurring within its specific geographic region. More formally, at a given time $t \in [0, T]$ in client $k$'s region, where $k = 1, 2, \ldots, K$, suppose that user $i$ has an interaction with item $j$, which can be represented as an edge $e_{k,t}^{(i,j)}$ in the graph. Client $k$ then has a local graph $\mathcal{G}_{k,t} = \{e_{k,0}, \ldots, e_{k,s}, \ldots, e_{k,t}\}$ at time $t$ including historical links, where we have suppressed the node indices on each link for simplicity. At any given time $t$, we include only those nodes $i, j$ in the graph that have at least one edge connecting them to another node at some time $s \leq t$. Note that $\mathcal{G}_{k,t}$ *grows over time* as more edges accumulate.

**Link prediction formulation.** For each client $k$ at time $T$, we have user set $\mathcal{I}_k = [I_k]$, item set $\mathcal{J}_k = [J_k]$, and local graph $\mathcal{G}_{k,T}$. During training, we initialized the trainable global user embedding layer $\boldsymbol{I} \in \mathbb{R}^{N \times d}$ and item embedding $\boldsymbol{J} \in \mathbb{R}^{M \times d}$, where $d$ is the dimension of the embedding vector. The goal of model training is to learn the user and item representations $\boldsymbol{\theta} = GNN(\boldsymbol{w}, \boldsymbol{I}, \boldsymbol{J}, \mathcal{G}_{k,T})$ by GNN model $\boldsymbol{w}$ and a predictor $\boldsymbol{\phi}$. The predictor $\boldsymbol{\phi}$ takes these learned representations as input to estimate the following probability:

$$\mathbb{P}\left( e_{k,T'}^{(i,j)} \in \mathcal{G}_{k,T'} | \boldsymbol{\theta}, \boldsymbol{\phi}; \mathcal{I}_k; \mathcal{J}_k; \mathcal{G}_{k,T} \right). \tag{1}$$

Here, $T$ represents the latest observed time in the training dataset, while $T'$ refers to a future time after $T$. Based on this probability we can then predict new links of nodes in the future. In order to define the relationship of $\boldsymbol{\phi}$ and $\boldsymbol{\theta}$, we can write the predicted probability as: $\mathbb{P}(e_{k,t}^{(i,j)}) = \boldsymbol{\phi}(s(\boldsymbol{\theta}_i, \boldsymbol{\theta}_j))$ where $s(\boldsymbol{\theta}_i, \boldsymbol{\theta}_j)$ is the cosine similarity between the user and item representations: $s(\boldsymbol{\theta}_i, \boldsymbol{\theta}_j) = \frac{\boldsymbol{\theta}_i \cdot \boldsymbol{\theta}_j}{\|\boldsymbol{\theta}_i\| \|\boldsymbol{\theta}_j\|}$. The GNN model typically consists of two or three layers (Zhang & Chen, 2018; Kipf & Welling, 2016; Yao et al., 2024a), and the learned user and item representations $\boldsymbol{\theta}$ are vector embeddings. The predictor $\boldsymbol{\phi}$ maps the cosine distance between user-item pairs to a probability score, which determines the likelihood of an interaction.

**Federated learning for link prediction.** Given the link prediction model above, we next explain how federated learning can be used to train this model. In the next subsection, we explain how FedLink builds

on this framework to handle the challenges of data heterogeneity, high computational overhead, and users moving from one client to another.

As usual in federated learning, training proceeds in rounds. Without loss of generality, suppose that training takes place at rounds $r = 1, 2, \ldots, R$; note that since we model edge arrivals as a continuous-time process, new edges may arrive in between the training rounds. At a specific round $r$ that takes place at time $t$, each client $k$ trains its local link prediction model on the local graph $\mathcal{G}_{k,t}$ with $L$ local stochastic gradient descent steps. Each client $k$ then sends its updated GNN model $\boldsymbol{w}_k^{(r)}$ and embedding layers $(\boldsymbol{I}_k^{(r)}, \boldsymbol{J}_k^{(r)})$ to the global server, which averages the received local models to update the global model $\boldsymbol{w}^{(r)}$ and global embedding layers $(\boldsymbol{I}^{(r)}, \boldsymbol{J}^{(r)})$. Once $R$ rounds of training have taken place, where $R$ can be chosen to ensure convergence, each client $k$ uses the trained model and its local graph $\mathcal{G}_{k,t}$ to predict the future links.

## 3.2 FedLink Algorithm

A principal challenge in the federated learning process described above is that it incurs **increasing computing overhead** over time as the graphs $\mathcal{G}_{k,t}$ grow with time $t$. Local graphs with millions of users and items, for example, may require excessive memory and runtime overhead during the local training process at each client. In addition to reducing this overhead, FedLink aims to adapt the federated learning process to solve the following three challenges:

- **Client Heterogeneity:** Clients may experience significantly different numbers of user-item interactions, e.g., if they are regions with different populations.

- **Temporal Heterogeneity:** The number of links arriving at each client may change over time. For example, users are more likely to engage with products during the day compared to the night. These patterns of user-item engagement may also vary by client, e.g., regions in different time zones.

- **Cross-client user nodes:** Users may move between clients, e.g., vacationing in different regions.

We use the main novel feature, *constant edge buffer* with *sharing of user embedding* to overcome these challenges. We outline the rationale, design, and effect of these two features below.

### 3.2.1 Constant Edge Buffer

Since links arrive at each region in a continuous manner, a naïve federated learning approach would trigger a model training round whenever a new link arrives. However, this may introduce prohibitive communication overhead due to communicating updated models with the central server whenever a new link arrives; moreover, clients with infrequent link arrivals would not be able to update their model whenever a new link arrives at a client with frequent link arrivals, as they may not yet have received any new links. Another naïve approach would be to trigger a new federated learning training round at regular time intervals, e.g., midnight GMT every day. Due to temporal and client heterogeneity, however, clients would then train on potentially very different numbers of newly arrived links, as shown in Figure 1 (bottom middle).

To handle this heterogeneity in link arrivals, we formalize an alternative approach where the link stream is partitioned into snapshots, which we call *buffers*, so as to maintain a *constant number of edges (links)* $C$ in each buffer. Formally, given a temporal network $\mathcal{G}$ (here we drop the client index $k$ for simplicity) with $m$ edges, let $\{e^1, e^2, e^3, \ldots, e^{m-1}, e^m\}$ be its edge stream. Construct a sequence of buffer graphs $\{\mathcal{G}^1, \mathcal{G}^2, \ldots, \mathcal{G}^s, \ldots, \mathcal{G}^S\}$ such that $\mathcal{G}^s = \{e^i | C(s-1) < i \leq Cs\}$, where $S$ is the index of the largest non-empty buffer. In our implementation, we use a First-In-First-Out (FIFO) strategy for managing these buffers, ensuring that the most recent edges displace the oldest ones when the buffer reaches capacity. Figure 1 (bottom right) illustrates the idea of a constant edge buffer.

Algorithm 1 illustrates the FedLink algorithm. As in traditional federated learning, in each training round $r = 1, 2, \ldots, R$, clients running FedLink first receive the global models $\boldsymbol{W}^{(r)}$ from the central server. Each client then computes $L$ local gradient steps with learning rate $\eta$. FedLink's difference from traditional federated learning on the fact that the client *chooses one of its buffers* to compute the gradient in each

---

**Algorithm 1** FedLink

---

Model parameters are represented by $\boldsymbol{W} = (\boldsymbol{w}, \boldsymbol{I}, \boldsymbol{J})$.
Each client $k$ maintains buffer graphs $\{\mathcal{G}_k^1, \mathcal{G}_k^2, \ldots, \mathcal{G}_k^S\}$, each with constant number of edges $C$.
**for** *round* $r = 1, \ldots, R$ **do**
    **for** *each client* $k \in [K]$ **do in parallel**
        Receive $\boldsymbol{W}^{(r)}$
        Set $\boldsymbol{W}_k^{(r,1)} = \boldsymbol{W}^{(r)}$
        **for** *local step* $l = 1, \ldots, L$ **do**
            Choose a buffer $\mathcal{G}_k^s \in \{\mathcal{G}_k^1, \mathcal{G}_k^2, \ldots, \mathcal{G}_k^S\}$
            Set $\boldsymbol{g}_{\boldsymbol{W}_k}^{(r,l)} = \nabla f_k(\boldsymbol{W}_k^{(r,l)}; \mathcal{G}_k^s)$
            $\boldsymbol{W}_k^{(r,l+1)} = \boldsymbol{W}_k^{(r,l)} - \eta \, \boldsymbol{g}_{\boldsymbol{W}_k}^{(r,l)}$ // `Update Parameters`
        **end**
        $\boldsymbol{\Delta}_{\boldsymbol{W}_k}^{(r,L)} = \boldsymbol{W}_k^{(r,L+1)} - \boldsymbol{W}_k^{(r,1)}$
        Send $\boldsymbol{\Delta}_{\boldsymbol{W}_k}^{(r,L)}$ to the server
    **end**
    // `Server Operations`
    $\boldsymbol{\Delta}_{\boldsymbol{W}}^{(r)} = \frac{1}{K} \sum_{k=1}^{K} \boldsymbol{\Delta}_{\boldsymbol{W}_k}^{(r,L)}$ // `Difference Aggregation`
    $\boldsymbol{W}^{(r+1)} = \boldsymbol{W}^{(r)} + \boldsymbol{\Delta}_{\boldsymbol{W}}^{(r)}$ and broadcast to local clients // `Update Global Models`
**end**
// `Perform Link Prediction`
**for** *each client* $k \in [K]$ **do in parallel**
    Predict future links using equation 1 based on $\boldsymbol{W}^{R+1}, \mathcal{G}_{k,T}$
**end**

---

local gradient step, instead of using the entire graph. The final gradient is then sent to the server, where aggregation of the client models takes place as usual.

This approach of sampling a constant-size buffer at each client *significantly reduces training overhead* compared to training on the full historical graph. Clients only need to store the links (and associated nodes) in the buffer in memory when training the federated learning model, instead of storing the entire historical graph. By sampling from past buffers, we also ensure that the model does not overfit to the most recent buffer, maintaining comparable model accuracy to using the entire historical graph.

Note that some users may move to another client while still having links in the buffer; while intuitively one might want to delete their associated links, the user may also return (e.g., if they temporarily travel to another country for vacation). We therefore update the buffer solely based on the age of the links and not the location of the user. More sophisticated methods might instead attempt to learn the relevance of each past link and update the buffer based on this estimated relevance; however, estimating link relevance to future predictions can be difficult in practice, so we leave this idea to future work.

### 3.2.2 Cross-Client User Embeddings

We next outline how FedLink generates link predictions when users move across clients. To do so, we take advantage of the fact that the link prediction mainly relies on the user and item embeddings ($\boldsymbol{I}, \boldsymbol{J}$), which is shared across clients. Thus, in order to generate link predictions for a user newly arrived at a client $k'$, we can *reuse the embedding $\boldsymbol{I}$ for this user that was learned at the user's previous client $k$.*

Since the items available at client $k'$ may differ from those at client $k$, users may not receive as accurate predictions after moving to client $k'$ as they might have at client $k$: their embedding was not trained to predict link formation for client $k'$'s items. However, we expect these predictions to be more accurate than those that would be generated if client $k'$ had to learn this user's embedding from scratch. We empirically validate this intuition in Section 4's evaluation.

### 3.3 Theoretical Analysis of FedLink

Since FedLink trains its GNN model on buffers of past data at each client, choosing the right *buffer size* may significantly impact the convergence of the training. While exactly quantifying the effects of buffer size

is difficult, due to the complexity of the GNN model, in this section we explore how changing the buffer size affects convergence. To do so, we use a well-known graph model called the dynamic stochastic block model (SBM) to model the evolution of the graph over time (Abbe, 2018; Keriven & Vaiter, 2022).

### 3.3.1 Dynamic Stochastic Block Model

For positive integers $K$ and $n$, a probability vector $p \in [0,1]^K$, and a symmetric connectivity matrix $B \in [0,1]^{K \times K}$, we define a static SBM as a random graph with $n$ nodes split into $K$ classes. The goal of a prediction method for the SBM is to correctly divide nodes into their corresponding classes, based on the graph structure. Each node is independently and randomly assigned a class in $\{1, ..., K\}$ according to the distribution $p$. Undirected edges are independently created between any pair of nodes in classes $i$ and $j$ with probability $B_{ij}$, which equals $\alpha$ if $i = j$ ($i$ and $j$ are in the same class) and $\mu\alpha$ if $i \neq j$ ($i$ and $j$ are in different classes), where $\alpha \in (0,1)$ and $\mu \in (0,1)$ are given parameters.

We consider a set of discrete time steps $t = 1, 2, \ldots, T$. At each time step $t$, the Dynamic SBM generates new intra- and inter-class edges according to the probabilities $\alpha$ and $\mu\alpha$ as defined for the SBM above. All edges persist over time. We assume a constant number of nodes $n$, number of classes $K$, and connectivity matrix $B$. Let $Y_t \in \{0,1\}^{n \times K}$ denote the matrix representing the nodes' class memberships at each time $t$, where $Y_{ik} = 1$ indicates that node $i$ belongs to the $k$-th class, and is 0 otherwise. We model changes in nodes' class memberships as a Markov process with a constant transition probability matrix $H \in [0,1]^{K \times K}$. Let $\varepsilon \in (0,1)$ denote the probability a node changes its membership. At each time step, node $v_i$ in class $j$ changes its membership to class $k$ with the following probability (independently from other nodes):

$$H_{j,k} = \mathbb{P}\left[Y_{ik}^t = 1 | Y_{ij}^{t-1} = 1\right] = \begin{cases} 1 - \varepsilon, & j = k \\ \dfrac{\varepsilon}{K-1}, & j \neq k, \end{cases}$$

While $\varepsilon$ may vary across classes $j$ in practice, for simplicity we suppose it is the same for each class.

We suppose that our learning task is to classify the nodes of the graph, i.e., to group nodes together so as to recover the membership matrix $Y$ up to column permutation at each time $t$. We expect link prediction to give similar convergence results, but as the analysis is more involved we present the node classification analysis for simplicity. We thus evaluate estimates $\hat{Y}$ of the membership matrix by defining the *relative error* of a classification estimate $\hat{Y}$ as

$$E(\hat{Y}, Y) = \min_{\pi \in \mathcal{P}} \|\hat{Y}\pi - Y\|_0, \tag{2}$$

where $\mathcal{P}$ is the set of all $K \times K$ permutation matrices and $\|.\|_0$ counts the number of non-zero elements of a matrix.

### 3.3.2 Class Behavior Over Time

We observe the behavior of class evolution over time by using the relative error function in (2) to characterize the change in classification over time, i.e. $E(Y_{t-\tau}, Y_t)$. Without loss of generality, we remove the permutation and keep class indices for columns of $Y$ constant for all membership matrices. Since node transitions are independent, the probability matrix $\mathbb{E}[Y_t|Y_{t-1}] = Y_{t-1}H^T$ allows us to find the expectation of the relative error between adjacent time steps. This is obtained by modeling the error as $n$ individual Markov chains between correct and incorrect classifications for each node. The probability of incorrectly predicting a node's class using the previous time step's information is $\varepsilon$ (the probability of class shifting between time steps). Therefore, $\mathbb{E}[E(Y_{t-1}, Y_t)] = 2\|Y_{t-1}\|_0\varepsilon = 2\|Y_t\|_0\varepsilon = 2n\varepsilon$, since every classification mistake increments the error metric by 2. For further time steps, due to the penalty function used in $E(\cdot, \cdot)$, we construct a two-state Markov chain for each user, where the states denote whether the user is in the same or different class as the current time. Then as $t \to \infty$, the system reaches a stationary distribution and $\mathbb{E}[E(Y_{t-\tau}, Y_t)] = 2n\frac{K-1}{K} \approx 2n$ for large $K$. We will approximate the error over time by the following continuous function:

$$E(t) = 2n - 2n(1 - \varepsilon)^{\eta t} \tag{3}$$

with an arbitrary convergence factor $\eta$ (Li & Orabona, 2019).

### 3.3.3 Buffer Error

We can now use Equation (3)'s bound on the convergence at time $t$ to derive the dependence on buffer size. At each client $k$, we model a link arrival rate of $\lambda_k$ i.i.d and with bounded variance across clients $\mathrm{Var}(1/\lambda) \leq \kappa^2$. Assuming relatively uniform arrivals, the buffer at a client will evenly cover a time period $\epsilon/\lambda_k$ with each link being representative of a current or outdated membership graph. As such, each link in the buffer also has a representative error dependent on how long in the past it has arrived at the client. We will approximate a summation over all representative errors of links present in the buffer of size $C$ by integrating $E(\tau)$ over the period $(0, C/\lambda_k)$ to determine a local *buffer classification error* for each client:

$$E_{buf}(C, \lambda_k) = \int_0^{C/\lambda_k} E(\tau)d\tau = 2nC/\lambda_k - \frac{(1-\varepsilon)^{\eta C/\lambda_k} - 1}{\eta \log(1-\varepsilon)} \tag{4}$$

Globally, the mean buffer classification error across all clients is: $E_{\text{classification}} = \frac{1}{K}\sum_k^K E_{buf}(\epsilon, \lambda_k)$. Simply analyzing the sum of the first terms of $E_{buf}$, we can observe a *linear relation between buffer size and classification error* due to data staleness: as we increase $C$, we incorporate more and more older data. While the second term of $E_{buf}$ also depends on $C$, its dependence on $C$ decays exponentially for large $C$, and the term overall is bounded from above by $\frac{1}{\eta \log(1-\epsilon)}$. Thus, we expect the effect of increasing $C$ to be dominated by the first term in $E_{buf}$. However, from existing analysis of federated learning such as Ye et al. (2023), it is also known that consistent dataset size across clients reduces variance on gradients and thus training error. Moreover, the rate of convergence $\eta$ is in general an increasing function of the buffer size, as larger buffers allow more data to be used in the training. Thus, when choosing the buffer size, we should be mindful of the tradeoffs between staleness effects and dataset size, as well as the increased memory and training time overhead caused by larger buffers. Experiments in the next subsection validate these tradeoffs.

### 3.3.4 Buffer Selection

Building upon our theoretical analysis of classification error and buffer size trade-offs, we conduct experiments to determine optimal buffer sizes specific to our dataset. Our experiments evaluate regions with populations varying from below 100,000 to over a million check-ins, with the US the largest country with 1,990,327 check-ins. We test buffer sizes ranging from 10,000 to 1,000,000 for the original data and from 10,000 to 500,000 for the 50% downsampled data, measuring the impact on AUC scores.

As shown in Figure 2, we observe a critical trade-off between buffer size, model performance, and GPU usage. The model's AUC shows a concave pattern as buffer size increases, which validates the trade-offs between buffer size and AUC Score. The GPU memory usage also increases with buffer size. Small buffer sizes are insufficient to capture enough historical information, especially for larger countries like the US, whose AUC continues to increase with larger buffer sizes. Larger buffer sizes exceed the historical data available in smaller regions, making them suboptimal and exhibiting approximately linear decreases for large buffer sizes, consistent with Equation (4). We use these identified optimal buffer ranges to proceed with further experiments.

## 4 Experiments

We finally validate FedLink on a real dataset that is naturally distributed across different geographical regions. We first describe the dataset, our baseline algorithms, and experiment settings before presenting our results. In order to assess the effectiveness of FedLink, we address the following research questions (RQs) to guide our experiments:

**RQ1. System Efficiency:** Does FedLink improve computational efficiency in terms of GPU memory usage and training time compared to baseline methods?

**RQ2. Model Accuracy:** Does FedLink's approach with a buffer method maintain accuracy comparable to traditional FL methods while addressing data heterogeneity across different regions?

**RQ3. Buffer & Data efficiency Analysis:** How do different components of FedLink (sharing user embeddings, buffer mechanisms, dataset size) contribute to its overall performance and efficiency?

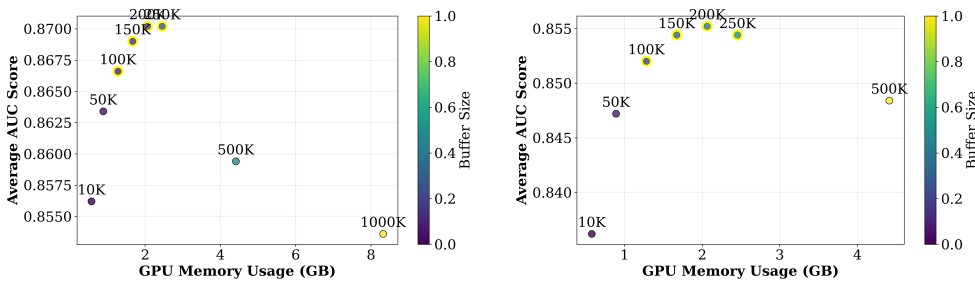

Figure 2: Impact of buffer size on AUC scores and GPU usage for (left) original data and (right) 50% downsampled data across different regions. The optimal buffer sizes that balances AUC scores and GPU usage are marked with yellow highlights.

### 4.1 Datasets

We use subsets of the Foursquare Global-scale Check-in Dataset (Yang et al., 2016) with User Social Networks[1] for our experiments. This dataset contains detailed global check-in data of users from 415 cities in 77 countries, with each city recording at least 10,000 check-ins between April 2012 and September 2013.

In our federated learning setup, each country serves as a separate client, where each check-in is represented as an edge connecting a user to a venue within that country. By using this extensive dataset, we select a representative subset of 10 countries for our experiments to evaluate the model's performance across different geographical regions and data size. The subset of the dataset we used for our experiment features the following statistics:

**Check-ins:** 7,705,646 check-ins made by 40,484 users across 1,273,946 venues.

**Population Distribution:** Selected countries show significant variation in user population sizes, reflecting real-world demographic differences, shown in Figure C.1.

**Check-in Distribution:** The check-in volumes also vary across countries, with the United States (US) having 1,990,327 check-ins, while Spain (ES) having 212,161 check-ins for example, shown in Figure C.2.

**Traveled user percentage:** Approximately 1.39% of users have check-ins across multiple countries, which we refer to as traveled users.

In addition to Foursquare, we also evaluate our approaches on the TGBL-Wiki(Huang et al., 2023)(Kumar et al., 2019) dataset from the Temporal Graph Benchmark (TGB). This dataset contains temporal user-page interactions from Wikipedia, represented as a heterogeneous graph. More detailed can be found in B.5.

**Interactions:** 157,474 timestamped interactions between 4,613 users and 4,614 pages, with timestamps ranging from 0 to 2,678,373.

### 4.2 Baselines

In order to evaluate the performance of our proposed method, we compare FedLink against several baseline models representing different federated training approaches and environment settings:

- **Local** (Hamilton et al., 2017): A Graph Neural Network (GNN) model that trained solely on full-batch local graph data without federated learning, demonstrating the baseline predictive power of GNN.

- **STFL** (Lou et al., 2021): A static GNN model added with a spatio-temporal federated learning framework to incorporate historical edge relationships, serving as a baseline for federated learning without buffer mechanisms.

- **4D-FED-GNN+** (Gürler & Rekik, 2022): A federated GNN model using a longitudinal approach to analyze temporal patterns within single-day edges.

---

[1]https://sites.google.com/site/yangdingqi/home/foursquare-dataset?pli=1

- **FedDGL** (Xie et al., 2024): A federated dynamic graph learning framework that addresses temporal evolution and data heterogeneity through global knowledge distillation and prototype-based regularization.
- **Feddy** (Jiang et al., 2022): A federated dynamic graph neural network that uses position prediction and secure aggregation to model temporal evolution, combining spatial graph convolution with temporal aggregation for dynamic federated learning.
- **FedLink**: Our proposed federated learning method, which employs edge buffers and includes the transmission of embeddings for users who have traveled.
- **FedLink-NoEmb**: FedLink, without sending embeddings for traveled users across clients.
- **FedLink-Local**: FedLink, using buffers for local training without any aggregation across clients. This serves as a control to understand the impact of buffer usage alone.
- **FedLink-MiniBatch**: FedLink that uses traditional random mini-batch sampling with the same batch size as the buffer, instead of our constant edge buffer approach.

### 4.3 Experiment Settings

Building upon our baseline methods, we evaluate FedLink's link prediction performance in our four subsets of countries in the Foursquare dataset. Based on the buffer size analysis in Section 3.3.4, we employ buffer sizes of 200,000 and 300,000, which demonstrated an optimal range for computational efficiency and model performance. In order to capture patterns of check-in behaviors and sufficient user travel across countries, we analyze the data over a 30-day period. The training settings for each model are as follows:

- **Local Training (Local and FedLink-Local)**: Both methods perform 60 local training iterations. Local uses the complete dataset, while FedLink-Local uses a single buffer per iteration.
- **Federated Training (STFL, 4D-Fed-GNN+, FedDGL, and Feddy)**: All methods perform 20 global training rounds with 3 local iterations per client. STFL, FedDGL and Feddy processes full historical data, while 4D-FED-GNN+ uses single-day data, representing scenarios with limited computational capacity. FedDGL selectively processes 10% of nodes as sensitive information for temporal knowledge distillation.
- **Federated Training with Buffer (FedLink, FedLink-NoEmb, and FedLink-MiniBatch)**: All methods perform 20 global training rounds with 3 local iterations per client, processing one buffer per iteration. FedLink shares user embeddings across clients for traveled users, while FedLink-NoEmb operates without this cross-client information sharing. FedLink-MiniBatch uses random sampling within buffers instead of FIFO sequential processing.

### 4.4 Experiment Results

**RQ 1: System Evaluation:** We evaluate FedLink's system efficiency across different country combinations by measuring training time, GPU memory usage, and AUC scores, as shown in Table 1. FedLink shows strong computational efficiency across all experimental settings. With a buffer size of 200,000, FedLink achieves comparable or faster training times to 4D-FED-GNN+ which only processes single-day data and FedDGL which process 20% of nodes for temporal knowledge. Moreover, FedLink also demonstrate its memory efficiency by requiring at least $3.41\times$ less GPU usage compared to full-batch methods like STFL and Feddy, while maintaining competitive AUC scores across all experiments. More detailed usage of system evaluation by country can be found in Table C.1.

**RQ 2: Model Accuracy Evaluation:** As shown in Table 1, FedLink achieves comparable AUC scores across all experimental settings, consistently ranking among the top two methods with STFL, Local, and FedDGL. Figure 4.4 further illustrates the efficiency-performance trade-off of FedLink, where its position in the upper-left corner indicates optimal balance: achieving high AUC scores while requiring significantly less GPU memory and training time. Further, the only method (4D-FED-GNN+) achieving similar efficiency shows substantially lower AUC scores. This result highlights that FedLink is the only method that can

Table 1: Performance comparison across six experimental settings. Measurements include training time (seconds), GPU memory usage (GB), and AUC scores. Results for Exp 1-4 averaged over 10 runs across a 30-day period using FedLink with buffer size 200,000 for 1,200 iterations. Exp 5-6 are TGBL-Wiki dataset results with single-run scores. **Bold** indicates best results, ***bold italics*** indicate second-best.

| **Exp 1** (US, BR, ID, TR, JP) | | | | **Exp 2** (MX, PH, ES, GB, IT) | | | |
|---|---|---|---|---|---|---|---|
| Methods | Time(s) | GPU(GB) | AUC | Methods | Time(s) | GPU(GB) | AUC |
| STFL | 2.406 | 6.003 | 0.870 | STFL | 2.082 | 1.220 | **0.848** |
| Local | 2.250 | 5.997 | **0.877** | Local | *1.867* | 1.220 | 0.847 |
| 4D-FED-GNN+ | 1.964 | *2.065* | 0.579 | 4D-FED-GNN+ | 1.857 | **0.894** | 0.567 |
| FedDGL | *1.932* | **1.892** | 0.873 | FedDGL | 1.950 | 0.933 | 0.832 |
| Feddy | 2.759 | 7.574 | 0.871 | Feddy | 2.389 | 2.386 | 0.841 |
| FedLink | **1.866** | *2.065* | *0.876* | FedLink | 1.997 | *0.894* | *0.848* |
| **Exp 3** (US, JP, BR, MX, ES) | | | | **Exp 4** (DE, NL, KR, FR, CA) | | | |
| Methods | Time(s) | GPU(GB) | AUC | Methods | Time(s) | GPU(GB) | AUC |
| STFL | 2.058 | 5.832 | 0.869 | STFL | 1.752 | 0.886 | 0.861 |
| Local | 1.889 | 5.812 | **0.876** | Local | 1.624 | 0.869 | 0.860 |
| 4D-FED-GNN+ | **1.649** | *1.789* | 0.587 | 4D-FED-GNN+ | *1.484* | *0.678* | 0.598 |
| FedDGL | 1.897 | **1.773** | *0.867* | FedDGL | 1.519 | 0.790 | *0.862* |
| Feddy | 2.254 | 6.184 | 0.871 | Feddy | 2.081 | 1.214 | 0.858 |
| FedLink | *1.782* | *1.789* | 0.871 | FedLink | **1.463** | **0.676** | **0.864** |
| **Exp 5** (TGBL-Wiki, Early Period) | | | | **Exp 6** (TGBL-Wiki, Later Period) | | | |
| Methods | Time(s) | GPU(MB) | AUC | Methods | Time(s) | GPU(MB) | AUC |
| STFL | 0.735 | 0.589 | 0.868 | STFL | 0.762 | 0.618 | *0.859* |
| Local | 0.711 | 0.589 | 0.866 | Local | 0.738 | 0.618 | 0.852 |
| 4D-FED-GNN+ | **0.582** | **0.303** | 0.824 | 4D-FED-GNN+ | **0.607** | **0.319** | 0.806 |
| FedDGL | 0.713 | 0.706 | *0.869* | FedDGL | 0.740 | 0.713 | 0.855 |
| Feddy | 1.185 | 0.943 | 0.864 | Feddy | 1.234 | 0.996 | 0.851 |
| FedLink | *0.617* | *0.365* | **0.875** | FedLink (Buffer) | *0.649* | *0.371* | **0.867** |

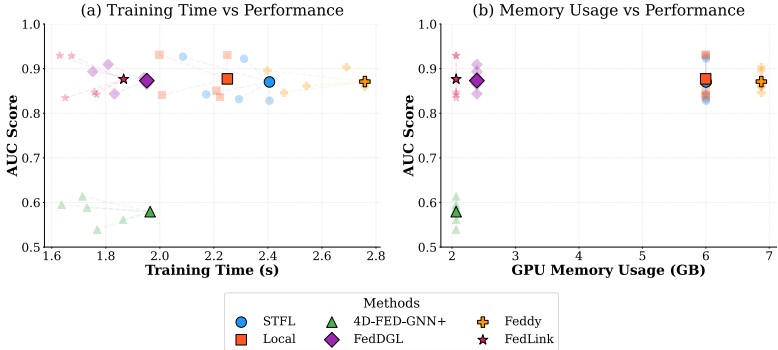

Figure 3: Comparison between different methods showing: (a) the relationship between training time and AUC Score, and (b) the relationship between GPU memory usage and AUC Score. FedLink optimally balances AUC scores with training time and memory usage across all methods.

balance efficiency with predictive performance.

**RQ 3: Ablation Study:** In order to evaluate the contribution of each component within FedLink, we conduct an ablation study comparing four variants: *Complete FedLink model* with buffer mechanism and cross-client embedding transmission; *FedLink-NoEmb*, which removes cross-client embedding for traveled users; *FedLink-Local*, which uses only local buffer training without federated learning; and *FedLink-Minibatch*, which replaces FIFO buffering with random mini-batch sampling. FedLink constantly achieves the highest AUC scores among all variants, demonstrating the effectiveness of both federated learning setting and FIFO buffer mechanisms, as shown in Table 2. The results of isolating only traveled users for different FedLink methods are in Table C.4.

Table 2: Ablation study by removing the FL part and the cross-client embedding part to test the effect on AUC Scores for different buffer sizes. Federated aggregation significantly contributes to FedLink's performance.

| Buffer Size = 200,000 | | | | | |
| --- | --- | --- | --- | --- | --- |
| Methods | US | BR | ID | TR | JP |
| FedLink (FL+Buffer) | **0.837±0.003** | 0.930±0.004 | **0.834±0.004** | **0.929±0.003** | **0.842±0.005** |
| FedLink-Local | 0.828±0.003 | 0.931±0.004 | 0.827±0.003 | 0.923±0.004 | 0.840±0.005 |
| FedLink-NoEmb | 0.832±0.003 | **0.931±0.006** | 0.829±0.006 | 0.931±0.005 | 0.836±0.007 |
| FedLink-MiniBatch | 0.823±0.005 | 0.926±0.004 | 0.825±0.005 | 0.921±0.005 | 0.834±0.006 |
| Buffer Size = 300,000 | | | | | |
| Methods | US | BR | ID | TR | JP |
| FedLink (FL+Buffer) | **0.848±0.003** | **0.940±0.004** | **0.906±0.004** | **0.966±0.003** | **0.950±0.005** |
| FedLink-Local | 0.832±0.003 | 0.926±0.006 | 0.890±0.006 | 0.963±0.005 | 0.946±0.007 |
| FedLink-NoEmb | 0.847±0.003 | 0.938±0.004 | 0.904±0.004 | 0.965±0.003 | 0.946±0.006 |
| FedLink-MiniBatch | 0.831±0.005 | 0.934±0.006 | 0.893±0.006 | 0.957±0.004 | 0.941±0.007 |

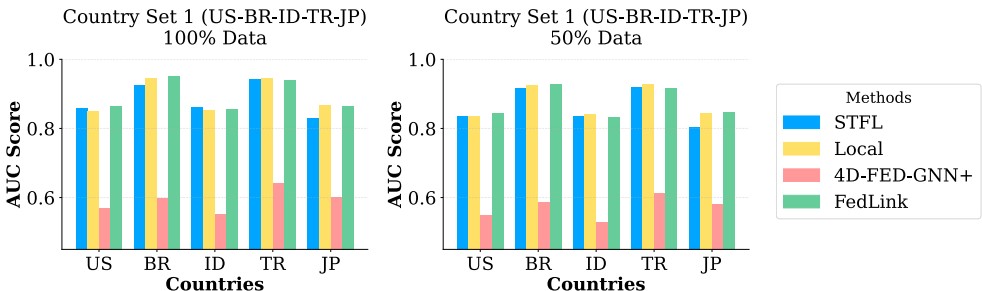

Figure 4: Performance comparison under data downsampling (100% vs 50%). Results demonstrate that FedLink maintains consistent AUC scores even with down-sampled data across both country sets. FedLink maintains comparable AUC scores with 50% data size across both country sets, while achieving reduced GPU memory consumption and faster training time.

To evaluate FedLink's performance under different data sizes, we conduct extensive downsampling experiments. Our experimental results demonstrate FedLink's resilience to data reduction, maintaining among highest accuracy with 50% downsampling of the training data shown in Figure C.5. Note that our experiments show that GPU memory scales proportionally with dataset size. This efficiency-performance balance is valuable for resource-constrained deployments. We also performed downsampling experiments for 50%, 25%, and 2% data size for different country combinations, which can be found in Table B.2.

## 5  Conclusion

In this paper, we propose FedLink, a federated link prediction algorithm for dynamic graphs. FedLink is motivated by the problem of recommending items, e.g., restaurants, to users in multiple regions. While it is desirable to train a model across regions to take full advantage of all available data, privacy constraints, and computational overhead may prohibit centralized training of a dynamic GNN model across all regions. FedLink addresses the challenges of computational overhead and privacy concerns in situations where graph data is localized, such as recommending restaurants to users in multiple countries. This approach has the added benefit of accommodating users who move across countries, by allowing them to simply share the user embedding across countries. Moreover, FedLink maintains edge buffers of fixed size at each client, thus alleviating the effects of temporal and inter-client heterogeneity in link arrivals over time. We show that FedLink significantly reduces memory requirements and improves training speed while matching the accuracy of centralized training. Future work includes extending FedLink to other dynamic graph applications and generalizing the standard federated learning convergence analysis to address the unique challenges of dynamic graph settings. Our methodology also has the potential to extend to time-series data analysis, node classification, and graph classification.

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

## 6 Appendix

## A Background

### A.1 Graph Convolutional Network

A multi-layer Graph Convolutional Network (GCN) Kipf & Welling (2016) with row normalization has the layer-wise propagation rule

$$H^{(l+1)} = \phi(\widetilde{D}^{-1}\widetilde{A}H^{(l)}W^{(l)}), \tag{5}$$

where $\widetilde{A} = A + I_N$, $I_N$ is the identity matrix, $\widetilde{D}_{ii} = \sum_j \widetilde{A}_{ij}$ and $W^{(l)}$ is a layer-specific trainable weight matrix. The activation function is $\phi$, typically ReLU (rectified linear units), with a softmax in the last layer for node classification. The node embedding matrix in the $l$-th layer is $H^{(l)} \in \mathbb{R}^{N \times D}$, which contains high-level representations of the graph nodes transformed from the initial features; $H^{(0)} = X$.

In general, for a GCN with $L$ layers of the form 5, the output for node $i$ will depend on neighbors up to $L$ steps away. We denote this set by $\mathcal{N}_i^L$ as $L$-hop neighbors of $i$. Based on this idea, the clients can first communicate the information of nodes. After the communication of information, we can then train the model.

## B Experiments

### B.1 Foursquare Dataset

We organize our experiments into four sets according to different country sizes as follows:
**Large Countries (EXP 1):** Top five countries with the highest check-in volumes: United States (US), Brazil (BR), Indonesia (ID), Turkey (TR), and Japan (JP) ;
**Midsized countries (EXP 2):** Five countries with relatively limited data: Mexico (MX), Philippines (PH), Spain(ES), UK(GB), and Italy(IT);
**Combinations (EXPs 3)**: Combinations of large and small countries (US, JP, BR, MX, ES) to assess FedLink's performance under client data heterogeneity;
**Small Countries (EXP 4)** Five small countries with size around 5% of the size of US: Germany(DE), Netherlands(NL), South Korea(KR), France(FR), and Canada(CA).

We present Fig C.1 to show the population distribution across the ten countries selected for our experiments. This population heterogeneity allow us to assess our model's ability to handle different levels of data density and user activity patterns.

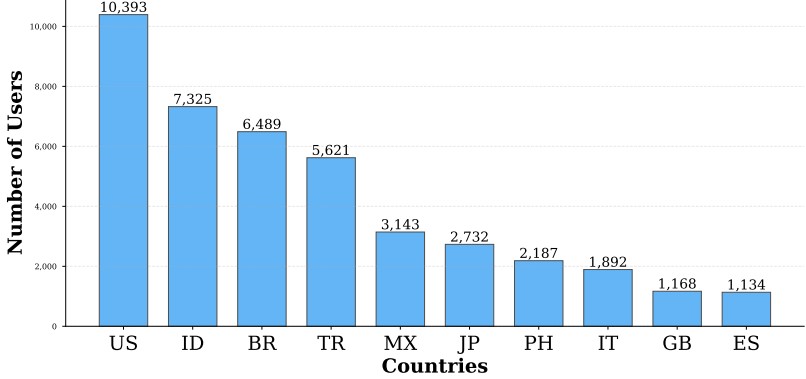

Figure C.1: Distribution of user population across selected 10 countries in the Foursquare dataset.

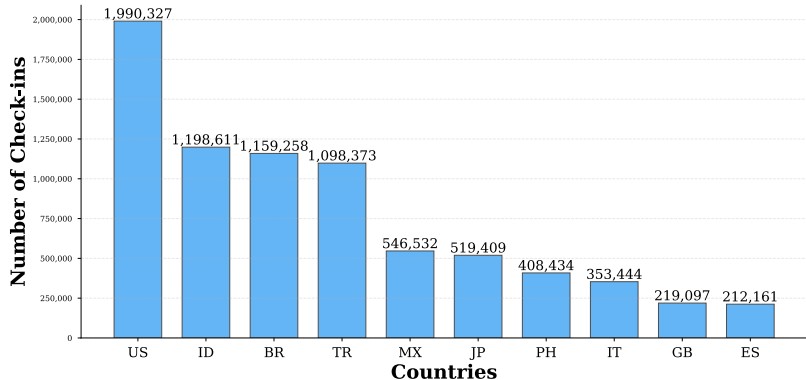

Figure C.2: Distribution of check-ins across selected 10 countries in the Foursquare dataset.

## B.2 Experimental Results

We present detailed performance metrics for different country combinations in Table C.1, showing training time for each country, total completion time, and GPU memory usage during training. The total training time per round is determined by the slowest client, as federated learning requires all clients to complete their local training before proceeding with global model updates. These detailed measurements demonstrate FedLink's computational efficiency compared to baseline methods. We also present detailed test AUCs for different experiments, showing AUC score for each country in Table C.2. To ensure model convergence,

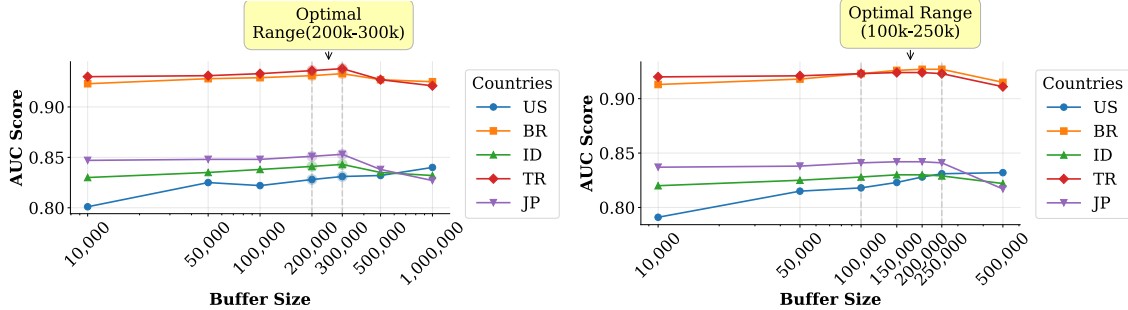

Figure C.3: Buffer Size Effect on AUC scores for five countries(US, BR, ID, TR, JP) on original data (left) and 50% downsampled data (right).

we monitor AUC scores over increasing training rounds. Our experiments use 180 global rounds, as we observe that model performance stabilizes well aorund this point, as shown in Figure C.4 for the convergence patterns.

## B.3 Buffer Selection

As shown in Fig C.3, we performed experiments on varying buffer size and evaluated its effect on model AUC and GPU usage. Detailed buffer size analysis on different countries are as follows:

1. For the US dataset, it is the largest country in number of check-ins, and is the only one that will benefit from increasing buffer size even beyond 500K with continuously improved AUC, as it haven't reach its full batch check-in size.

2. For medium-sized countries (BR, TR, MX), the improvement plateaus around 200K, and actual performance decreases beyond 500K.

Table C.1: Training Time of FedLink with buffer size 200,000 on clients for 1200 Iterations. Each result represents the total training time, averaged over 10 runs. FedLink consistently trains faster and has lower GPU memory usage, than all other algorithms except 4D-FED-GNN+ which has a much lower AUC.

| | US | BR | ID | TR | JP | Total Time (s) | Training GPU (GB) |
|---|---|---|---|---|---|---|---|
| STFL | 2.172 | 2.085 | 2.293 | 2.312 | 2.406 | 2.406 | 5.997 |
| Local | 2.009 | 1.998 | 2.223 | 2.250 | 2.210 | 2.250 | 5.997 |
| 4D-FED-GNN+ | 1.864 | 1.637 | 1.768 | 1.714 | 1.730 | *1.864* | **2.065** |
| FedLink | 1.758 | 1.630 | 1.650 | 1.673 | 1.766 | **1.766** | **2.065** |
| | MX | PH | ES | GB | IT | Total Time (s) | Training GPU (GB) |
| STFL | 1.688 | 2.082 | 1.755 | 1.624 | 1.551 | 2.082 | 1.220 |
| Local | 1.540 | 1.867 | 1.580 | 1.577 | 1.549 | *1.867* | 1.220 |
| 4D-FED-GNN+ | 1.493 | 1.857 | 1.488 | 1.481 | 1.480 | **1.857** | **0.894** |
| FedLink | 1.583 | 1.996 | 1.602 | 1.689 | 1.690 | 1.996 | *0.894* |
| | US | BR | JP | MX | ES | Total Time (s) | Training GPU (GB) |
| STFL | 2.058 | 1.881 | 1.711 | 1.866 | 1.994 | 2.058 | 5.832 |
| Local | 1.798 | 1.837 | 1.650 | 1.889 | 1.809 | 1.889 | 5.830 |
| 4D-FED-GNN+ | 1.606 | 1.601 | 1.594 | 1.649 | 1.583 | **1.649** | *1.789* |
| FedLink | 1.648 | 1.630 | 1.630 | 1.782 | 1.781 | *1.782* | **1.789** |
| | US | IT | GB | TR | ES | Total Time(s) | Training GPU(GB) |
| STFL | 2.015 | 1.894 | 1.723 | 1.879 | 1.982 | 2.015 | 4.449 |
| Local | 1.806 | 1.860 | 1.662 | 1.867 | 1.814 | 1.860 | 4.447 |
| 4D-FED-GNN+ | 1.650 | 1.614 | 1.607 | 1.643 | 1.596 | **1.650** | *1.302* |
| FedLink | 1.554 | 1.627 | 1.624 | 1.632 | 1.639 | **1.639** | **1.301** |
| | DE | NL | KR | FR | CA | Total Time(s) | Training GPU(GB) |
| STFL | 1.552 | 1.684 | 1.752 | 1.651 | 1.696 | 1.752 | 0.886 |
| Local | 1.487 | 1.624 | 1.583 | 1.598 | 1.554 | 1.624 | 0.886 |
| 4D-FED-GNN+ | 1.393 | 1.401 | 1.484 | 1.422 | 1.379 | *1.484* | *0.678* |
| FedLink | 1.347 | 1.408 | 1.395 | 1.463 | 1.421 | **1.463** | **0.676** |

Table C.2: Test AUCs on different test sets with varied countries combination. Each result is tested over a 30-day period with a 200,000 buffer size. The result is averaged over 10 runs. FedLink consistently achieves the best (bold) or second-best (bold italics) AUC compared with local training and STFL which require significantly more training time.

|  | US | BR | ID | TR | JP |
|---|---|---|---|---|---|
| STFL | *0.842±0.004* | 0.927±0.006 | 0.832±0.006 | 0.922±0.004 | 0.828±0.008 |
| Local | 0.841±0.008 | **0.931±0.007** | **0.836±0.009** | **0.931±0.006** | **0.851±0.008** |
| 4D-FED-GNN+ | 0.561±0.006 | 0.595±0.007 | 0.539±0.012 | 0.614±0.004 | 0.588±0.014 |
| FedLink | **0.847±0.003** | *0.930±0.004* | *0.834±0.004* | *0.929±0.003* | *0.842±0.005* |
|  | MX | PH | ES | GB | IT |
| STFL | *0.876±0.004* | **0.865±0.004** | *0.839±0.003* | **0.824±0.004** | 0.836±0.004 |
| Local | 0.873±0.006 | *0.865±0.009* | 0.838±0.009 | *0.822±0.010* | *0.837±0.010* |
| 4D-FED-GNN+ | 0.640±0.002 | 0.544±0.002 | 0.546±0.002 | 0.568±0.002 | 0.536±0.001 |
| FedLink | **0.877±0.004** | 0.865±0.003 | **0.839±0.005** | 0.821±0.006 | **0.837±0.005** |
|  | US | BR | JP | MX | ES |
| STFL | *0.831±0.003* | *0.929±0.006* | 0.832±0.008 | 0.899±0.005 | 0.853±0.008 |
| Local | **0.838±0.008** | **0.931±0.007** | **0.843±0.009** | **0.908±0.006** | **0.865±0.008** |
| 4D-FED-GNN+ | 0.560±0.012 | 0.594±0.013 | 0.590±0.012 | 0.646±0.012 | 0.547±0.014 |
| FedLink | 0.828±0.005 | 0.929±0.001 | *0.835±0.004* | *0.903±0.005* | *0.859±0.003* |
|  | US | IT | GB | TR | ES |
| STFL | *0.842±0.005* | 0.853±0.007 | *0.872±0.004* | 0.871±0.003 | 0.854±0.009 |
| Local | **0.843±0.010** | **0.864±0.007** | 0.864±0.008 | **0.879±0.004** | **0.866±0.007** |
| 4D-FED-GNN+ | 0.552±0.007 | 0.563±0.009 | 0.571±0.011 | 0.583±0.008 | 0.547±0.013 |
| FedLink | 0.840±0.006 | *0.861±0.003* | **0.873±0.007** | *0.874±0.003* | *0.863±0.004* |

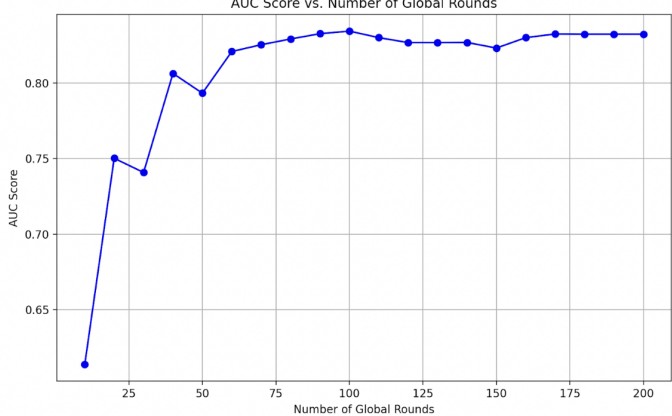

Figure C.4: AUC scores convergence analysis for FedLink over global training rounds

3. For smaller countries (JP, GB, ES), and the downsampled version of medium sized countries in **Figure C.3(right)**, optimal performance is achieved at even smaller buffer sizes between 100K-150K.

## B.4  Ablation Study

We also present detailed experimental results for ablation study for downsampling data. Table B.2 presents model performance under various data reduction scenarios (50%, 25%, and 2% of original data) across different country combinations.

Table C.3: Test AUCs on down-sampled test sets with varied countries combination with 50%, 25%, 2% size reduction on different country combination's data input. Each result is tested over a 30-day period with a 200,000 buffer size. The result is averaged over 10 runs. FedLink consistently achieves the best (bold) or second-best (bold italics) AUC compared with local training and STFL which require significantly more training time.

| | US(50%) | BR(50%) | ID(50%) | TR(50%) | JP(50%) |
|---|---|---|---|---|---|
| STFL | *0.836±0.004* | 0.915±0.003 | 0.835±0.006 | 0.920±0.004 | 0.804±0.007 |
| Local | 0.835±0.008 | **0.926±0.008** | **0.8412±0.008** | **0.928±0.005** | **0.843±0.007** |
| 4D-FED-GNN+ | 0.549±0.007 | 0.586±0.008 | 0.529±0.010 | 0.613±0.003 | 0.579±0.010 |
| FedLink | **0.843±0.004** | *0.928±0.004* | *0.833±0.003* | *0.917±0.003* | *0.847±0.005* |
| | US(50%) | BR(50%) | JP(50%) | MX(50%) | ES(50%) |
| STFL | *0.831±0.004* | *0.929±0.006* | 0.832±0.006 | 0.899±0.005 | 0.853±0.008 |
| Local | **0.838±0.008** | **0.931±0.007** | **0.843±0.009** | **0.908±0.006** | **0.865±0.008** |
| 4D-FED-GNN+ | 0.560±0.012 | 0.594±0.013 | 0.590±0.012 | 0.646±0.012 | 0.547±0.014 |
| FedLink | 0.828±0.005 | 0.929±0.001 | *0.835±0.004* | *0.903±0.005* | *0.859±0.003* |
| | US(25%) | BR(25%) | JP(25%) | MX(25%) | ES(25%) |
| STFL | 0.683±0.006 | 0.834±0.006 | 0.720±0.005 | 0.780±0.008 | 0.701±0.006 |
| Local | **0.687±0.008** | **0.835±0.007** | **0.735±0.009** | **0.798±0.006** | **0.713±0.008** |
| 4D-FED-GNN+ | 0.508±0.012 | 0.519±0.013 | 0.538±0.012 | 0.546±0.012 | 0.509±0.014 |
| FedLink | **0.693±0.005** | **0.837±0.001** | *0.731±0.004* | *0.793±0.005* | *0.709±0.003* |
| | US(2%) | BR(2%) | JP(2%) | MX(2%) | ES(2%) |
| STFL | *0.831±0.004* | *0.929±0.006* | 0.832±0.006 | 0.899±0.005 | 0.853±0.008 |
| Local | **0.838±0.008** | **0.931±0.007** | **0.843±0.009** | **0.908±0.006** | **0.865±0.008** |
| 4D-FED-GNN+ | 0.560±0.012 | 0.594±0.013 | 0.590±0.012 | 0.646±0.012 | 0.547±0.014 |
| FedLink | 0.828±0.005 | 0.929±0.001 | *0.835±0.004* | *0.903±0.005* | *0.859±0.003* |
| Method | Full Data | 50% Large | 50% Small | 25% Data | 2% Data |
| STFL | 0.913±0.004 | 0.859±0.005 | 0.869±0.006 | 0.744±0.006 | 0.869±0.006 |
| Local | **0.915±0.008** | **0.870±0.009** | **0.754±0.008** | **0.877±0.008** | **0.877±0.008** |
| 4D-FED-GNN+ | 0.567±0.009 | 0.566±0.009 | 0.587±0.013 | 0.509±0.014 | 0.549±0.010 |
| FedLink | *0.916±0.003* | *0.874±0.004* | *0.871±0.004* | *0.802±0.004* | *0.889±0.007* |

Table C.4: Test AUCs for only traveled users under the three ablation study methods of FedLink, FedLink-Local and FedLink-NoEmb.

| | US | BR | ID | TR | JP |
|---|---|---|---|---|---|
| FedLink (FL+Buffer) | **0.808±0.001** | **1.000±0.001** | **0.964±0.036** | **0.833±0.001** | **0.923±0.182** |
| FedLink-Local | 0.552±0.002 | 0.732±0.158 | 0.548±0.124 | 0.531±0.004 | 0.610±0.215 |
| FedLink-NoEmb | 0.759±0.003 | 0.724±0.086 | 0.813±0.108 | 0.783±0.164 | 0.811±0.007 |

Table C.4 specifically focuses on traveled users, comparing performance across three variants: FedLink, FedLink-Local, and FedLink-NoEmb. This analysis shows the importance of both federated learning and embedding sharing mechanisms for predicting traveled users. FedLink consistently outperforms other variants on traveled users, demonstrating the effectiveness of cross-client information sharing for users who move between regions.

### B.5 TGBL-Wiki

The dataset of TGBL-Wiki includes 157474 edges with timestamps from 0 to 2678373. There are total of 4613 users and 4614 items, with 133892 user-item interactions.

To evaluate temporal performance across different time periods, we partition the TGBL-Wiki dataset into two temporal subsets based on timestamp ordering: an **earlier period** containing the first half of interactions

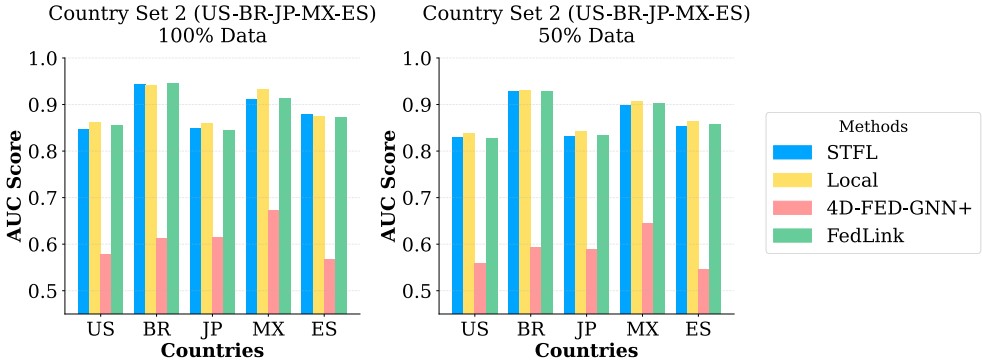

Figure C.5: Performance comparison under data downsampling (100% vs 50%). Results demonstrate that FedLink maintains consistent AUC scores even with down-sampled data across both country sets. FedLink maintains comparable AUC scores with 50% data size across both country sets, while achieving reduced GPU memory consumption and faster training time.

(timestamps 0 to 1339186) and a **later period** containing the second half (timestamps 1339187 to 2678373). This temporal split allows us to assess how federated learning methods perform on older versus newer interaction patterns, providing insights into temporal generalization capabilities.

# C   Example Use Cases

We evaluate our novel training method on the Foursquare dataset, demonstrating its effectiveness in real-world location-based service scenarios.

## C.1   Link Prediction on Tabular Data

Most user data in Adobe is stored in tabular format, where each row in the table represents information on a user information (e.g., user IP address, location, website, and purchased product). User information is also updated with time. However, entries in this table may be missing, e.g., due to faulty data collection. By modeling the user table information as a dynamic graph, we can perform link prediction on the tabular data and fill in the missing part of the table, allowing us to more easily use it for various tasks (e.g., purchase behavior prediction).

## C.2   Website Behavior Prediction

Predicting websites to be visited by users allows dynamic pre-caching of the site content, reducing communication costs and response latency. By modeling visits as a user-website bipartite graph, we can predict frequently visited websites. Such visit behavior is cyclic and dynamic, requiring regular updates and on-time predictions.

## C.3   User Identity Verification

Identity verification helps to validate users' product subscriptions. By modeling users and their behavior as nodes in a dynamic graph, we can detect anomalous behavior and classify nodes as "good" users and "bad" actors.

