# OpenReview forum: "Federated Link Prediction on Dynamic Graphs"
_TMLR — Rejected by TMLR_

### Review · Reviewer_M9fx · 2025-04-28

**Summary Of Contributions:**

The submission introduces FedLink, a federated learning framework for link prediction on dynamic graphs, using constant edge buffers to reduce memory usage and training time while maintaining accuracy. It incorporates cross-client embedding sharing for users moving across regions and provides a theoretical analysis of buffer size trade-offs. Experiments on the Foursquare dataset show FedLink’s efficiency and ability to handle data heterogeneity.

**Audience:**

Yes

**Claims And Evidence:**

Yes

**Requested Changes:**

- Explicitly describe the buffer update strategy (e.g., "oldest edges are replaced first") in Section 3.2.1. The impact of buffer update policies (e.g., FIFO vs. relevance-based) appears to be unexplored.
- A convergence analysis or generalization bounds for FedLink's training dynamics would strengthen the theoretical contributions of this paper.

**Strengths And Weaknesses:**

**Strengths​​:**
- ​​The paper tackles a timely and practical challenge: training dynamic GNNs under privacy constraints and computational efficiency demands. The focus on link prediction in scenarios like user-item recommendations is well-motivated, with clear real-world applications.
- The use of fixed-size buffers to store historical edges at each client is appears to be a novel approach to mitigate computational overhead and data heterogeneity. This balances staleness and dataset size while reducing memory usage.
- Extensive experiments on the Foursquare dataset demonstrate FedLink’s efficiency (3.41x memory reduction, 28.9% faster training) and competitive accuracy (matching centralized models). Ablation studies confirm the importance of federated aggregation and embedding sharing.

**Weaknesses​​:**
- The dynamic SBM analysis, while useful, oversimplifies real-world graph dynamics. A formal convergence analysis or generalization bounds for FedLink could strengthen theoretical claims.
- The mechanism for updating buffers (e.g., FIFO, random, or relevance-based replacement) is not quite clear. This could impact performance in practice.

---

### Review · Reviewer_h74s · 2025-04-28

**Summary Of Contributions:**

The paper focuses on the problem of link prediction on dynamic graphs from the federated learning perspective. Although dynamic graphs are useful to capture temporal information, data localization causes some issues, for instance, a centralized model cannot be used due to data protection regulations. To solve such issues, the authors propose a novel method called FedLink. FedLink is claimed to be memory efficient and computes faster than the compared methods.

**Audience:**

Yes

**Broader Impact Concerns:**

The authors clearly discuss the broader impact of the proposed work in Section 5.

**Claims And Evidence:**

Yes

**Requested Changes:**

All the concerns mentioned above should be addressed to improve the paper.

Although Section 3.2.2 provides a nice take on users moving across clients, a user may disappear on a client without moving to another one, or a new user may emerge. An experiment regarding such case can demonstrate method's effectiveness in a better way.

In addition, the disappearance of a user invalidates the some of the stored edges in the buffer since that user does not exist. How does the proposed method handle such cases (i.e. disappearance of one of the nodes linked via a stored edge)?

In addition to performance analysis regarding the buffer size, an ablation study concerning the GPU memory usage may be beneficial.

**Strengths And Weaknesses:**

**Strengths**

The authors propose a federated link prediction framework for dynamic graphs, claiming that they are the first ones to tackling the data imbalance issue among the clients.

A detailed theoretical analysis is provided regarding the proposed model and class evolution behaviour.

An ablation study is applied to the proposed method to demonstrate the effects of each module.

The paper is well written and easy to follow.

**Weaknesses**

Only 2 baseline studies are given. A benchmark study on temporal graphs is cited but none of the methods are considered for baselines. This should be expressed. In addition, the Feddy method is also cited in Related Work section but is not considered for baselines.

The buffer size is determined as 200k and 300k. However, on Figure 2, the given plot does not justify using a 20 times larger buffer than the smallest buffer, as the performance difference does not seem to be dramatic. The model only experiences a dramatic increase in US check-in data. A performance analysis regarding the buffer size is required.

Only Foursquare dataset is used. Experiments on different datasets such as tgbn-genre might demonstrate the proposed model's performance on link prediction better.

Omission of a reference to the from prior work in the literature [1].

[1] Yao, Y., Li, Y., Fan, X., Li, J., Liu, K., Jin, W., Ravi, S., Yu, P.S. and Joe-Wong, C., 2024. FedGraph: A Research Library and Benchmark for Federated Graph Learning. arXiv preprint arXiv:2410.06340.

---

### Review · Reviewer_QsJp · 2025-05-01

**Summary Of Contributions:**

This paper studied the problem of federated link prediction in the context of dynamic graphs. It introduced a FedLink framework by designing data buffers to control the training efficiency and sharing the user embedding across clients. The experimental results demonstrated the improved computational efficiency of the proposed framework.

**Audience:**

Yes

**Claims And Evidence:**

No

**Requested Changes:**

(1) The technical novelties of the proposed framework can be further clarified and verified.

(2) Section 3.1 discusses a cosine distance between the user and item representations. But the formula $d(\theta_i, \theta_j)$ is given by the cosine similarity.

(3) The extension of the provided theoretical analysis to link prediction tasks can be further clarified.

(4) More recent baselines, e.g., [1], can be included to validate the effectiveness of the proposed model over state-of-the-art techniques.

[1] Zaipeng Xie, Li Likun, Xiangbin Chen, Hao Yu, and Qian Huang. "FedDGL: Federated Dynamic Graph Learning for Temporal Evolution and Data Heterogeneity." In The 16th Asian Conference on Machine Learning (Conference Track). 2024.

**Strengths And Weaknesses:**

Strengths:

(S1) A novel FedLink framework is proposed for federated link prediction on dynamic graphs.

(S2) The impact of data buffer size is analyzed theoretically and empirically.

(S3) The experimental results confirm the computational efficiency of the proposed framework.

Weaknesses:

(1) The technical novelties of the proposed framework are unclear. There are two crucial components: constant edge buffer and cross-client user embedding.
- The buffer selection process is very similar to mini-batch training in traditional GNNs. For each local step, only a subset of edges/nodes will be selected to update model parameters. Though it constructs the buffers/mini-batches based on the time information, it is unclear regarding the benefits of edge buffers compared to traditional mini-batch training.
- Following the previous concern, the trade-off involving buffer size is also similar to the trade-off induced by local batch sizes commonly seen in standard federated learning under data heterogeneity.
- Section 3.2.2 suggests that the user embedding can be reused across clients. However, this idea will introduce additional data privacy concerns. Why is this user embedding allowed to be shared? How will it affect the privacy protection policy in federated learning?

(2) The theoretical analysis in Section 3.3 is hard to follow.
- One major concern is that the analysis is built on a node classification task, instead of the link prediction tasks studied in this paper. It mentioned that "we expect link prediction to give similar convergence results, but as the analysis is more involved we present the node classification analysis for simplicity". It is unclear why similar convergence results can be expected for link prediction tasks.
- The estimation of $\mathbb{E}[E(Y_{t-1}, Y_t)]$ and $E(t)$ in Eq. (3) is confusing. First, how are these two terms calculated? More detailed information can be provided. Second, will $E(t)$ converge to $2n$ when $t \to \infty$, regardless of the value of $\eta$? If so, does it imply that the classification error will always accumulate over time?
- The "linear relation between buffer size and classification error" is only observed from the first term of Eq. (7). It is not convincing, as both terms of Eq. (4) involve the buffer size $C$.
- It states that "the rate of convergence $\eta$ is in general an increasing function of the buffer size, as larger buffers allow more data to be used in the training". It is unclear how this conclusion is obtained.

(3) The performance improvement of FedLink is marginal, especially compared to the Local method.
- "Local" is the simplest baseline for federated learning. Since it doesn't need to share parameters, it is much better than FL methods in privacy protection. This implies that if FL methods achieve worse or comparable performance than "Local", it is more convincing to use "Local" method to reduce communication costs and privacy leakage risks.
- Besides, the proposed buffer strategy can also be applied to "Local" to further reduce the running time. The baseline "Local + Buffer" can be considered in the experiments.

---

### Decision · Action_Editor_QQHa · 2025-06-14

**Recommendation:** Reject

**Additional Comments:**

NA

**Audience:**

Yes

**Audience Explanation:**

Yes, some audiences would be interested in knowing the findings.

**Claims And Evidence:**

No

**Claims Explanation:**

While the paper presents a timely and practically relevant framework for federated link prediction on dynamic graphs, it falls short in terms of theoretical rigor and clarity of novelty. Two reviewers raise substantial concerns regarding the insufficient justification for the extension of the theoretical analysis from node classification to link prediction, the lack of convergence or generalization guarantees, and the oversimplified use of dynamic SBM. Although the authors provide extensive rebuttals and additional experiments, these efforts do not fully resolve the fundamental theoretical concerns or convincingly clarify the technical contributions over standard mini-batch and local learning strategies. Given the remaining ambiguity and limited gains over stronger baselines, I recommend rejection at this time, and encourage the authors to further strengthen the theoretical foundation and empirical justification in a future revision.

**Resubmission Of Major Revision:**

The authors may consider submitting a major revision at a later time.